# The Effect of Anthocyanins from *Dioscorea alata* L. on Antioxidant Properties of Perinatal Hainan Black Goats and Its Possible Mechanism in the Mammary Gland

**DOI:** 10.3390/ani12233320

**Published:** 2022-11-28

**Authors:** Yuanxin Zhang, Huiyu Shi, Yanhong Yun, Haibo Feng, Xuemei Wang

**Affiliations:** College of Animal Science and Technology, Hainan University, Haikou 570228, China

**Keywords:** perinatal, Hainan black goats, *Dioscorea alata* L. anthocyanin, oxidative damage, pre-protection

## Abstract

**Simple Summary:**

As they are the primary meat goats of Hainan, how to safely and stably transport Hainan black goats through the perinatal period has become one of the major issues. Likewise, anthocyanin-rich *Dioscorea alata* L. is grown to some extent in Hainan but is wasted in great quantities. The aim of this paper is to explore the effect of *Dioscorea alata* L. anthocyanins (DAC) on the antioxidant capacity of perinatal Hainan black goats, as well as the possible mechanism on the effect of prior protection, thereby enhancing the reproductive ability of goats and developing characteristic forage resources. Our results indicated feeding a diet in which *Dioscorea alata* L. powder replaces 30% of corn can increase the antioxidant capacity of perinatal female goats and newborn kids. Furthermore, under H_2_O_2_ stress, the use of DAC to treat goat mammary epithelial cells (GMEC) may serve an effective pre-protective function. This will provide data to support the protection of perinatal black goat females and the development of specific food resources.

**Abstract:**

(1) Background: The mammary glands of the perinatal goats are susceptible to reactive oxygen species (ROS) leading to oxidative injury. Although *Dioscorea alata* L. is rich in anthocyanins with high safety and excellent free-radical-scavenging ability, the effect and mechanism of *Dioscorea alata* L. anthocyanins (DAC) on the antioxidant capacity of the black Hainan goat has been the subject of few studies to date; (2) Methods: For this reason, feeding experiments were performed by feeding experimental diets, and the pre-protective capacity of DAC on goat mammary epithelial cells was explored on the basis of the established model of H_2_O_2_ injury; (3) Results: As well as altering rumen fermentation parameters in perinatal female goats, dietary challenge also improves antioxidant capacity in their blood and milk. thereby enhancing children’s antioxidant capacity and increasing their resistance to oxidative stress. However, we also found that DAC pretreatment was capable of activating both Nrf2 and MAPK/JNK pathways, which results in enhanced antioxidase activity and elimination of ROS; (4) Conclusions: Together, these findings suggest that DAC may have a pre-protective role on perinatal Hainan black goats through the regulation of Nrf2 and MAPK/JNK pathways in GMEC.

## 1. Introduction

Ruminants in the perinatal period are typically in a negative energy balance (NEB) state as a result of severe physiological changes such as desiccation, childbirth, and lactation [1,2]. Goats in this condition are subject to metabolic and infectious diseases, which is one of the main issues on the farm [3]. A growing numbers of studies have shown that when the body is in the NEB state, it needs to burn a lot of fat for energy [4,5]. Lipid peroxidation, however, will result in a large number of ROS. This results in the damage of large cellular molecules, such as DNA and proteins, which can result in oxidative stress [6,7]. Activation of NF-κB, MAPK/JNK and other inflammatory signaling pathways can be induced by a high concentration of ROS [8,9,10], leading to an increased incidence of mastitis, severely compromising maternal and child health [11,12,13].

*Dioscorea alata* L. was planted on a large scale in Hainan. In addition, because of its rich anthocyanins, it has been confirmed by our research group that it can alleviate colitis in mice, and that it plays a pre-protective role in the oxidative damage of IPEC-J2 cells [14,15]. Anthocyanins are well-recognized as potent natural antioxidants that inhibit lipid peroxidation and scavenge free radicals [16,17]. In addition, anthocyanins from different sources can increase the body’s antioxidant capacity in various ways. For example, the Nrf2 signaling pathway was activated to positively regulate the mRNA expression levels of HO-1, CAT, GSH-Px, and other antioxidant enzymes [18,19,20] or to inhibit the NF-κB and MAPK/P38 signaling pathways, resulting in reduced expression of pro-inflammatory factors and attenuation of inflammation [21,22,23].

In contrast, there is a paucity of studies investigating the effects of *Dioscorea alata* L. anthocyanin (DAC) on antioxidant capacity in Hainan black goats during the perinatal period, and it remains to be determined whether DAC can protect mammary epithelial cells (GMEC) of goats from oxidative stress. The aim of the present study was, therefore, to investigate the effect of DAC in feeding experiments and to assess its ability to pre-protect on GMEC and its possible molecular mechanisms.

## 2. Materials and Methods

### 2.1. Materials

DAC was extracted and purified by the animal Nutrition and Feed Science research team [15], College of Animal Science and Technology, Hainan University. The purity of the product was 39.59%, and the major active ingredients were the following: procyanidin B2 (1852.98 ng/mg), procyanidin B4 (475.24 ng/mg), rutin (352.08 ng/mg), and centaurin-3-galactoside (212.96 ng/mg). Freshly prepared *Dioscorea alata* L. (grown for 4 months) was washed with water, air-dried, sliced, and dried in an air-blown drying oven at 60 °C constant temperature to constant weight and stored away from the light. A feed mixer was used to physically grind and stir the resulting drying samples together with feedstock from goats’ diets. The Hainan black goats were supplied by the Hainan Black Goat Breeding Base at Danzhou Campus at Hainan University.

Fetal bovine serum (FBS) was purchased from HyClone (HyClone, Logan, UT, USA). The 0.25% trypsin-edta, DMEM/F12 and penicillin–streptomycin mixture (100×) were purchased from Gibco (Gibco, New York, NY, USA). Epidermal growth factor was purchased from Sigma (MERCK, Kenilworth, NJ, USA). RIPA, PMSF, PBS, ROS, LDH, and BCA kits were purchased from Beyotime (Shanghai, China). CAT, SOD, T-AOC, GSH-Px, and MDA kits were purchased from Nanjing Jiancheng Bioengineering Institute (Nanjing, China). CCK-8, DMSO, reverse transcription kit (BL699A), SYBR green qPCR mix (BL698A), and RNase free dd-H_2_O, ECL chemiluminescence substrate were purchased from Biosharp (Hefei, China). RNA-easy isolation reagent (R701) was purchased from Vasyme (Nanjing, China).

### 2.2. Reagent Preparation

Complete medium: DMEM/F12 + 6% FBS + 1% penicillin–streptomycin mixture + 0.1% EGF; termination medium: DMEM/F12 + 10% FBS; and cryopreservation liquid: 90% FBS + 10% DMSO were used.

DAC working solution: Dissolve DAC in DMEM/F12 medium and adjust the concentration to 1 mg/mL. Filter DAC with 0.22 μm filter for sterilization and then serve as DAC original solution for reserve. DMEM/F12 medium (containing 1% double antibody) was used to dilute the original DAC solution to 0, 10, 15, 20, 30, 40, 50, 60, 70, 80, 90, and 100 μg/mL as the working solution.

### 2.3. Breeding and Management of Hainan Black Goats

A total of 28 nonpregnant Hainan Black Goats were selected from the breeding base for estrus synchronization and artificial insemination. These goats had an average age, weight, and parity of 2.4 ± 0.4 years, 36.7 ± 3.1 kg, and 2.3 ± 0.7 times, respectively. The mean of body condition score (BCS) was 3.4 ± 0.6, based on the Wang’s study about goats [24]. All goats were ear-tagged prior to experimentation and had unified deworming and Brucella detection. Following artificial insemination, the goats were randomly divided into control and treatment groups with 14 goats per group. A total mixed diet was used for the control group, and a 30% corn diet was substituted for *Dioscorea alata* L. powder (experimental feed for short) for the treatment group. The goats were fed twice a day at 08:00 and 16:00, and each goat was fed 500g concentrate daily. In the current study, the basal diet consisted of concentrate: roughage (fresh King grass) = 1:1 (dry matter). Table 1 and Table 2 show the specific nutrient composition and feed formula, and these nutritional needs are in line with the Nutritional Requirements of Meat-Type Sheep and Goat (NY/T 816-2004, China). Artificial insemination was then terminated as Day 1, and the formal trial lasted from 60 d to 120 d, during which time water was freely consumed.

### 2.4. Determination of Antioxidant Properties

Blood samples were collected at 1, 2, 3, 7, 14, and 21 d postpartum prior to the morning feed, and milk samples were collected at 3 h post-morning feed. Ruminal fluid was collected by oral intubation before feeding on the 14 d of the experimental period and on the 21 d in the morning after delivery. Immediately after filtration, pH was measured using four layers of gauze, and then sent to the Chinese Academy of Agricultural Sciences for determination of the main fermentation parameters. Blood samples from pups at 7, 14, and 21 d after birth were collected in a similar manner to protect the health of neonates. Antioxidant capacity was determined in the serum and milk of goats and the serum of children, respectively. Simultaneously, ruminal fluid fermentation parameters were measured.

### 2.5. Cell Culture

GMEC was purchased from iCellbioscience (Shanghai, China, iCell-0016a). The cells were cultured in a T-25 tissue culture flask at 37 °C in an incubator with 5 % CO_2_, which is housed in the cell culture room, College of Animal Science and Technology, as well as Hainan University. Cells were passaged every 36 h, and logarithmic growth-phase GMEC were harvested for experimental treatment at 24 h. All cells were treated with drugs and placed in incubators.

### 2.6. Measurement of Cell Relative Survival Rate

Following extraction, cells were seeded in 96-well plates with 3.8 × 10^3^ cells per well and placed in an incubator for 24 h. A total of 48 treatment groups were set up with 6 biological replicates in each treatment group. A 12 × 4 two-factor fully randomized trial design was adopted; namely, the first factor was the concentration of DAC: 0 (control), 10, 15, 20, 30, 40, 50, 60, 70, 80, 90, and 100 μg/mL, and the second factor was the processing time of the DAC: 3, 5, 7, 9 h. The amount of 100 μL of the working solution was added to each well for the assay. The procedure was carried out according to the instructions of the CCK-8 kit, and the concentration and time of DAC treatment were chosen on the basis of the relative survival rate. 

### 2.7. Measurement of LDH Activity and the Relative Content of ROS

A preliminary determination of the concentration and processing time of DAC with improved relative cell survival rate was performed by cell relative survival rate significantly. After extraction, 24 treatment groups were set up with 3 replicates in each treatment group, and the cells were plated in 6-well plates with 1.3 × 10^5^ cells in each well and placed in an incubator for a further 24 h. The amount of 3 mL of working solution was added to each well for testing. The supernatant was aspirated after culturing up to each time point, and LDH activity in the supernatant was determined as per kit instructions. The remainder of the cells were cleaned with PBS for 3 times, and the relative amount of ROS in the cells was determined according to the instructions.

### 2.8. Measurement of Antioxidant Ability

Likewise, SOD and CAT activities as well as T-AOC in the supernatant were determined as directed. The remainder of the cells were washed 3 times with PBS followed by the addition of 500 μL RIPA and 5 μL PMSF to each well. The wells were lysed for 30 min in an ice bath, and then the bottom of the well was scraped off with a cell scraper, and the lysing solution was collected in a 1.5 mL centrifuge tube for labeling followed by centrifugation at 1000 r/min for 5 min. The amount of MDA and the activity of GSH-Px in the supernatant after centrifugation were determined according to kit instructions.

### 2.9. Measurement of Pre-Protection Capacity

On the basis of the results above, the optimal C_1_ concentration and T_1_ time of DAC treatment were chosen for the pre-protection trial. The concentration T_2_ of VC with the same clearance rate as that of the DAC selected was then determined by the DPPH scavenging assay as the positive group. As a result, cells were randomly divided into 6 groups for treatment: control group, H_2_O_2_ group, DAC group, VC group, DAC pre-protective group, and VC pre-protective group. Of these, the C_2_ and T_2_ treatment conditions of the H_2_O_2_ group have been proven by preliminary testing [27]. DAC and VC were also added first to the pre-protection treatment group to address T_1_–T_2_, and then H_2_O_2_ was added to address T_2_ together. The pre-shielding capability of DAC was similarly investigated by measuring the same indices mentioned above.

### 2.10. RNA Extraction and qPCR

The cells were then processed, and the original medium was discarded and slowly cleaned with PBS three times. Total RNA was extracted from the cells using the RNA-Easy protein extraction kit. Reverse transcription was subsequently performed using the reverse transcription kit, and the resulting cDNA product could be used directly for qPCR. Table 3 shows the primer sequences for the genes to be tested.

### 2.11. Western Blotting Analysis

Total protein was extracted following treatment of cells using RIPA. The protein concentration of each sample was determined using the BCA protein concentration kit as per instructions. To keep the volume of protein loading the same, the dilution was calculated, which was mixed with 5 × SDS loading buffer in a 4:1 ratio and then denaturated in a boiling water bath for 3–5 min for Western blot. Primary antibodies against were P-JNK (1:1000 dilution, 4688, Cell Signaling, Danvers, MA, USA), JNK (1:1000 dilution, 9258, Cell Signaling), Cytochrome C (1:200 dilution, AC909, Beyotime, Shanghai, China), and Caspase-3 (1:1000 dilution, AC030, Beyotime), respectively. The secondary antibodies of goat anti-rabbit IgG (1:1000 dilution, A0208, Beyotime) and goat anti-mouse IgG (1:1000 dilution, A0216, Beyotime) were used to combine the primary antibodies. β-Actin (1:1000 dilution, 4970, Cell Signaling) was used as the internal reference. Quantified densitometric analysis was conducted using a ChemiDoc MP (BIO-RAD, Hercules, CA, USA), and the gray value was analyzed by using Image-J software (ORACLE, Redwood, CA, USA).

### 2.12. Statistical Analysis

All data were analyzed by one-way ANOVA using SPSS 24 statistical software (IBM, AL, NY, USA). The significance of differences within groups was compared by Duncan’s method, and the significance of differences between groups was compared by independent sample *T*-test. The results were expressed as mean ± standard error, and a difference was considered significant with *p*-value < 0.05. Excel was used to generate linear regression curves. Finally, the GraphPad Prism 8 software is used for graph analysis.

## 3. Results

### 3.1. The Effects of Experimental Feed on Antioxidant Capacity of Perinatal Hainan Black Goats and Newborn Kids

#### 3.1.1. The Effect of Experimental Feed on Blood and Milk Antioxidant Capacity of Female Goats

The results in Table 4 show that with the passage of time, GSH-Px activity first increased and then decreased over time, but the change in MDA content was the opposite. In addition, when the two groups were compared, with the exception of CAT, the remaining indicators showed significant changes. We then further explored this effect in milk, as well as the significantly increased milk antioxidant capacity in the experimental group. Table 5 shows these results. These results indicate an apparent effect of DAC on goats.

#### 3.1.2. The Effect of Experimental Feed on Fermentation Parameters in Rumen of Perinatal Female Goats

As can be seen from the results of Table 6, due to the goats’ special digestive structure, changing the feed can easily affect the fermentation status of the ruminal microorganisms and, thus, alter fermentation parameters. At the same time, there were different changes between the two groups, except for acetic acid concentration. In addition to feeding, delivery also had an effect on the rumen, especially propionic acid. 

#### 3.1.3. The Effect of Experimental Feed on Antioxidant Capacity in Blood of Newborn Kids

Lastly, the antioxidant capacity of the child’s blood varied according to the influence of the goats in Table 7. Influenced by milk, the kids in the treatment group had better antioxidant levels in their blood.

### 3.2. The Optimum Effect Condition of DAC on GMEC

#### 3.2.1. The Effect of DAC on Relative Survival Rate of GMEC

We focus this effect on the GMEC on the basis of previous work. This experiment was designed to facilitate the exploration of DAC pre-shielding capability by first exploring the optimum effect conditions of DAC on GMEC. As can be seen in Table 8, with the prolongation of the treatment time, the relative survival rate of the GEMC showed an increasing trend. In particular, for 7 h, 50 μg/mL and 9 h, 20 μg/mL, the relative survival rates were as high as 118.8 ± 2.9% and 114.9 ± 4.6%, respectively.

#### 3.2.2. The Effect of DAC on LDH Activity and ROS Relative Content

As can be seen from Table 9, LDH activity in the culture medium increased first and then tended to be stable with the increase of DAC treatment time, while the change trend of ROS relative content was fittingly opposite. In addition, the effect of high concentration treatment in the same time point is more significant.

#### 3.2.3. The Effect of DAC on Antioxidant Capacity of GMEC

According to Table 10, Table 11, Table 12, Table 13, Table 14 and Table 15, as expected, the activities of CAT, SOD, GSH-Px, and T-AOC in some treatment groups increased significantly at 7 or 9 h, whereas the ROS and MDA content was opposite (*p* < 0.05). These suggest that DAC can improve the antioxidant capacity of GMEC. On the basis of the comprehensive analysis of the previous determination results, it can be concluded that the optimal condition of DAC treatment for GMEC was 20 μg/mL for 9 h.

We also analyzed the main effect and mode of mutual effect of all indicators under both time and concentration variables. Table 16 shows that T-AOC was primarily affected by time (*p* < 0.05). MDA content was affected mainly by concentration (*p* < 0.05). CAT activity was affected by both time and concentration (*p* < 0.05). The relative survival rate, activities of SOD, GSH-Px, LDH, and relative contents of ROS were affected by both treatment time and concentration (*p* < 0.05) and the interaction of the two factors (*p* < 0.05).

### 3.3. The Effect of Pre-Protection and Mechanism of DAC on H_2_O_2_-Induced GMEC Oxidative Damage

#### 3.3.1. Selection of VC Concentration

The first step was to determine the concentration of positive VC by the rate of DPPH clearance. As shown in Figure 1, the regression equation between VC concentration and the rate of DPPH clearance is Y = 1.7787X − 1.3648, R^2^ = 0.9894, (n = 6). The DPPH clearance rate of 20 μg/mL DAC was 22.434 ± 0.596%, and the VC concentration calculated by regression equation was 13.380 μg/mL for subsequent tests.

#### 3.3.2. The Effect of Pre-Protection from DAC on Antioxidant Capacity of GMEC

In addition, DAC pre-shielding capability was assessed by measuring the same indices mentioned above. It is clear from Figure 2 that DAC may protect the H_2_O_2_-induced damage to the GMEC antioxidant system in advance, and the effect is significant. Despite the higher LDH activity, relative ROS concentration, and MDA content in the DAC pre-protected group compared with the control group, they were also lower than those in the impaired group (*p* < 0.05). Conversely, with the exception of T-AOC, CAT, SOD, and GSH-Px, activities were greater than those in the control group, and were all greater than those in the impaired group (*p* < 0.05).

#### 3.3.3. The Effect of Pre-Protection from DAC on Key Genes of Nrf2 and MAPK-JNK Signaling Pathway in GMEC

Previous findings have shown that DAC can promote GMEC proliferation within a certain concentration range, enhance its antioxidant capacity, and efficiently scavenge intracellular free radicals and lipid peroxidation products. Thus, Nrf2 and MAPK-JNK, the major functions of which are related to antioxidant and apoptosis, were chosen as the objects in this study in order to explore their possible regulatory mechanisms. Figure 3 shows that the expression levels of the genes Nrf2, HO-1, and NQO-1 in the DAC pre-protected group were higher than those of the control group, in particular HO-1 (*p* < 0.05). It is interesting to note that the expressions of two inflammatory factors, IL-1β and TNF-α, were also significantly elevated, especially IL-1β (*p* < 0.05). This was accompanied by an increase in the expression of apoptosis-related Bax/Bcl-2, Caspase-3, and Caspase-9 (*p* < 0.05).

#### 3.3.4. The Effect of Pre-Protection from DAC on Key Proteins of MAPK-JNK Signaling Pathway in GMEC

Due to the unexpected observations, subsequent protein expression was further investigated in this study. Figure 4 shows that JNK was more strongly phosphorylated in the impaired group (*p* < 0.05). Analogously, the expressions of Cytochrome-C and Caspase-3 are also increased (*p* < 0.05). This is in stark contrast to the findings of other studies, suggesting that DAC can affect GMEC in various ways. Original western blot figures and data are shown in Appendix A.

## 4. Discussion

To explore its effect on practical application, we performed animal experiments. If the body is in the NEB state, this will not only increase the burden of childbirth on the body but will also have a negative impact on the yield and quality of colostrum. The goats’ bodies were found to be significantly affected by the birthing process. Mainly reflected in the activities of T-AOC, SOD and GSH-Px in blood of female goats in the control group were lower than those in the treatment group 1 d after delivery, while the content of MDA was higher. This stimulation leads to an increased incidence of mastitis. Studies have shown that the content of ROS, LPO, and NOx in milk during mastitis significantly increases [28], which results in a decrease in the antioxidant capacity of the milk. On this basis, we found that the changes in antioxidant capacity in the milk of control goats were the same as those in the blood at 1 d postpartum. Together, these suggest that oxidative stress is not confined to the mammary gland but to the entire body. Rather, the treatment group did a good job of resistance. There was also an increase in antioxidant capacity from 2 to 7 d postpartum in the blood and milk of goats in the control group. This may be due to the body gradually adapting to the current state and gradually recovering as lactation continues. As time progresses, however, this ability diminishes and oxidative stress becomes more severe. The kids’ ongoing development may increase the demand for milk, thereby increasing lactational burden in goats on the basis of NEB. In fact, this was not the case for goats in the treatment group, indicating that the experimental feed not only enhanced the antioxidant capacity of goats but also enhanced their fitness. Similarly, Tsiplakou supplemented sheep feed with methionine or choline and betaine protected from the rumen, which improved GST activity in plasma and plasma ferric reducing ability (FRAP) [29].

The rumen is the primary site for microbial synthesis of bacterial proteins and VFA in ruminant animals, which is affected by feed [30] composition and immune status of the body [31]. Results showed that the ruminal fermentation parameters of the female goats in both groups had the same pattern of change during delivery, and all indices were less than those of the prenatal goats except for the ratio of acetic acid to propionic acid. This may be because the goats’ immune status was also altered during the birthing process, and the number and types of substrates for microbial fermentation were altered in the NEB condition, resulting in a weakened and impaired fermentation process. Previous studies have shown that microbial abundance in the rumen of postpartum dairy cows changes [32], and acetic acid and propionic acid, as the major VFA, require different substrates in order to produce [33]. Therefore, in this study, the mode of fermentation after delivery tended to be acetic acid, the TVFA concentration decreased as a result of feeding on *Dioscorea alata* L., and the acetic acid: propionic acid ratio and pH value were raised. It has been documented that *Dioscorea alata* L. grown for 4 months has a starch content of 60% [34], which can result in its starch content being 10% lower than maize (70%) [35], which can lead to changes in fermentation profiles. As Guo pointed out, a high starch content was capable of reducing the ratio of acetic acid to propionic acid in the rumen of Hu sheep [36]. Anthocyanins can still remain active in the rumen [37] and, thus, display antibacterial properties in addition to their antioxidant properties. There was a significant reduction in TVFA at postnatal Day 21 in the treatment group, suggesting that DAC feeding over a long period inhibits the bacterium’s ability to digest fibre. Lazalde also found in his studies of the ruminant that the anthocyanidin of *Hibiscus sabdariffa* L. had the same impact on the rumen [38] and an increase in ruminal pH caused by the decrease in the ruminal content of TVFA, which can be easily explained. We will, of course, continue to emphasize this point in future experiments.

During oxidative stress in goats, the redox status of the placental tissue that is responsible for the transfer of nutrients between the female and the young was also vulnerable to destruction. Animal placentas have been found to have higher levels of glutathione S transferase (GST), which is responsible for the reduction of oxidative stress prevention, to maintain redox homeostasis in pregnancy [39]. However, oxidative stress can increase ROS content and reduce GST level, thus affecting the state of the fetus at birth [40]. Coupled with changes in the antioxidant capacity of the goats’ blood and milk, the oxidative stress status of goats at birth was found to negatively affect antioxidant capacity in children. However, supplementation of goats with *Dioscorea alata* L. powder in the feeding process may modify this negative effect. These results indicate that the antioxidant capacity of children can be regulated by altering the level of maternal oxidative stress. However, because of the lack of samples collected at 1, 2, and 3 days postpartum, it is not possible to determine whether the source of the influence is from the maternal placenta or milk, so further research is required.

In line with the main effect analysis, we did indeed find an interesting outcome. The activity of LDH in GMEC increased with increasing treatment concentration and duration, in particular 100 μg/mL, indicating that the addition of LDH would lead to damage of GMEC’s membranes. However, DAC can also enhance the relative rate of cell survival and enhance GMEC proliferation. Therefore, this study hypothesizes that DAC has two effects on cells within a certain time frame: on the one hand it destroys the cell membrane structure and results in cell injury; however; it also promotes cell proliferation and results in the number of new cells exceeding the number of damaged cells, resulting in relative proliferation. This may indicate that DAC induces apoptosis within a period of time but simultaneously shortens the cell cycle, leading to more new cells than apoptotic ones. These results are in agreement with Xie’s findings that daidzein can increase the proportion of G2-M phase in bovine mammary epithelial cells and reduce the proportion of S phase in an attempt to promote cellular proliferation [41]. Our further studies found that DAC could increase the activities of CAT, SOD, and GSH-Px, increase T-AOC, and reduce the relative content of ROS, thus leading to the decrease of MDA content. However, CAT and SOD activities were lower than the control group at 3 h and then gradually increased, indicating that DAC addition inhibited or reduced synthesis of both enzymes. Combined with changes in LDH activity and relative rates of proliferation, this further enhances the possibility of the above analysis. Even though GSH-Px activity did not show the same changes at 3 hours, it was also decreased to varying degrees at 3 to 5 h and increased again at 5 h, which may be due to the delayed response of related mechanisms of gene regulation in different antioxidant signaling pathways, which requires further investigation.

The integrity of the cell membrane system is maintained by VC, which can protect antioxidant enzymes within the cell, scavenge free radicals, and reduce lipid peroxidation and its products, thereby protecting the cell membrane from damage by oxidative stress [42]. Therefore, VC was selected as a positive control to explore DAC’s antioxidant mechanism laterally. According to the results, compared with the control and the impaired, the DAC pre-protected group can well reduce the relative content of ROS by improving the activities of CAT, SOD, and GSH-Px, thus reducing the generation of MDA, proving that DAC can perform the same role as VC to good effect. Interestingly, however, VC was not able to improve CAT activity, and the increase in SOD activity was much less than DAC. VC is known to be able to combine directly with hydrogen peroxide and its free radicals to perform an antioxidant role [43]; therefore, enhanced CAT activity is not required to degrade H_2_O_2_. Differentially, DAC may increase CAT transcription and expression by increasing Nrf2 expression [44], so DAC may improve CAT activity better than VC. But VC may still increase SOD and GSH-Px mRNA levels in mice [45] and can also positively regulate TRAIL expression in human breast cancer cells and activate Caspase [46]. It turns out that compared with the DAC pre-protection group, the VC pre-protection group did not show any change in the mRNA expression levels of TNF-α and Caspase-9, whereas expression levels of other genes were altered to varying degrees, indicating that the mechanism of VC pre-protection regulation was not the same as DAC.

Further testing revealed that HO-1 mRNA expression was strongly upregulated in the lesion condition, while NQO-1 expression was in the opposite direction. These results were consistent with the above observations that CAT activity and T-AOC were transiently enhanced. A temporary decrease in MDA content was observed, as well as a slow increase in relative ROS content during the construction of damage models [27]. We confirmed that under H_2_O_2_ stimulation, cells could increase antioxidant enzyme activity by upregulating HO-1 expression and reducing Bax/Bcl-2 to resist activation of the apoptosis pathway. In contrast, limited HO-1 alone is unable to withstand persistent and stable oxidative stress, leading to dramatically increased expression of IL-1β and TNF-α. This promotes JNK phosphorylation, activation of mitochondrial apoptosis pathways, and the release of Cytochrome C, and, ultimately, apoptosis. Unexpectedly, the expression of genes encoding Caspase-3, protein, and Caspase-9 were not significantly increased in the injured group, suggesting that they may not be causative, although Caspase-8 and Caspase-7 may, on the hand, be causative [47].

We noted that DAC had a bilateral effect on cells during GMEC treatment with DAC, which could be tested by changes in genes and proteins. There was a significant increase in the mRNA expression levels of IL-1β and TNF-α following DAC treatment, especially the former, indicating that DAC as a foreign additive was stimulating the cells to produce inflammatory cytokines as well as inhibiting both HO-1 and NQO-1 expression. However, the upregulation of Nrf2 expression indicated that DAC was beginning to enhance the antioxidant capacity of the cells through activation of the Nrf2 signaling pathway, giving rise to the above results, while activating the JNK phosphorylation pathway through the abundant expression of inflammatory factors. This promotes the release of Cytochrome C and ultimately increases the expression of the gene and protein for Caspase-3, leading to apoptosis. This is in agreement with Feng [48] and Ma [49], although the presence of apoptosis did not result in reduced cell numbers; this further substantiates the above speculation that DAC may shorten the cell cycle. Therefore, the expression of IL-1β and TNF-α mRNA was significantly increased in response to both H_2_O_2_ and DAC, thereby inducing apoptosis. Because of DAC’s function, the gene expression levels of Nrf2, HO-1, and NQO-1 were also greatly increased and reached inter-group peak under pre-protection treatment. These results confirm Cellat’s finding that pretreatment with *Smilax excelsa* L. anthocyanidins can increase the antioxidant capacity of rat testis tissue through activation of the Nrf2 /HO-1 pathway [50]. All in all, one could further hypothesize that DAC plays a pre-protective role against oxidative stress in the following ways: Activation of the Nrf2 signaling pathway may ameliorate the activity of GMEC antioxidant enzymes and induce apoptosis of damaged cells via the MAPK/JNK pathway over time. This may reduce secondary damage caused by the leaked contents of damaged cells after disruption, as well as shorten the cell cycle to promote proliferation in order for new cells to participate again in the mechanism of antioxidant regulation.

## 5. Conclusions

The diets fed with dried *Dioscorea alata* L. powder instead of 30% corn can significantly improve the antioxidant capacity of Hainan black goats and their kids during the perinatal period. 

The relative survival rate and antioxidant capacity of the cells can be effectively improved by treating GMEC cells with a 20 ug/mL DAC for 9 h. In addition, this has suggested that DAC activates two mitochondria-dependent signaling pathways, MAPK/JNK and Nrf2. The combined action of both signaling pathways reduced ROS content, thereby contributing to the effect of DAC pre-protection on H_2_O_2_ induced GMEC oxidative damage. Figure 5 illustrates this well.

## Figures and Tables

**Figure 1 animals-12-03320-f001:**
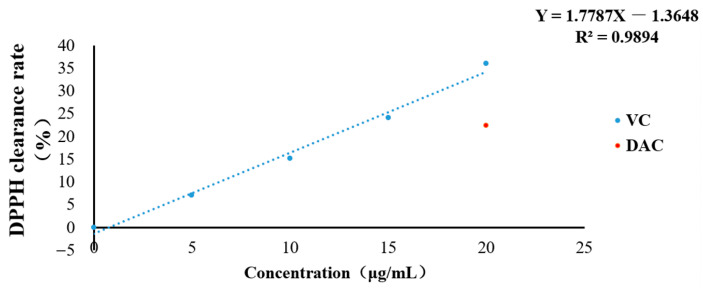
The clearance rate of DPPH.

**Figure 2 animals-12-03320-f002:**
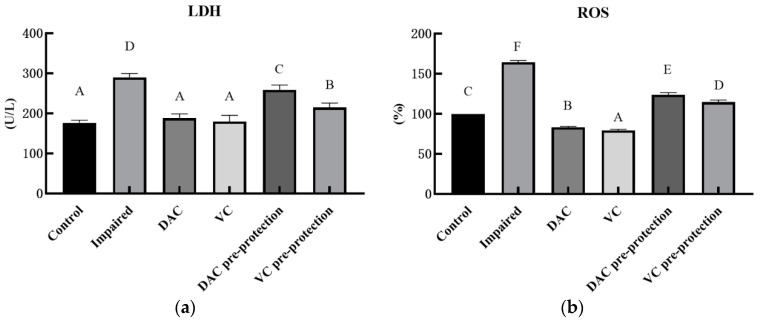
Effect of DAC pre-protection on antioxidant capacity of GMEC. The main indicators reflecting the antioxidant capacity of cells include: (**a**) LDH activity, (**b**) ROS relative content, (**c**) CAT activity, (**d**) T-AOC, (**e**) SOD activity, (**f**) GSH-Px activity, and (**g**) MDA contents. Compared with the control group, in the graph, values with different capital letter superscripts mean significant difference (*p* < 0.05), while with the same letter superscripts mean no significant difference (*p* > 0.05). The same as below of the figure.

**Figure 3 animals-12-03320-f003:**
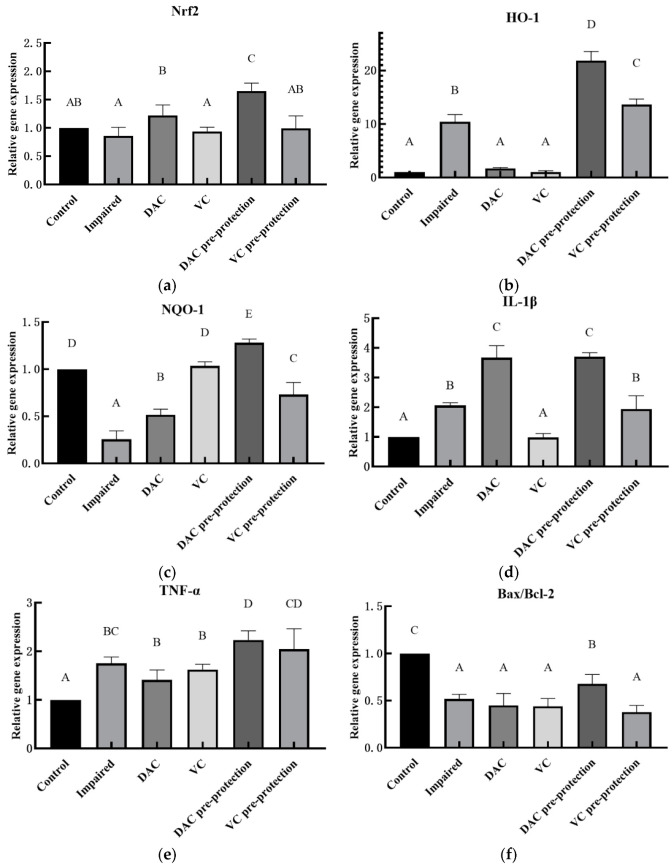
Effect of DAC pre-protection on mRNA relative expression of key genes in Nrf2 and MAPK-JNK signaling pathway in GMEC. Key genes in the Nrf2 pathway for antioxidant function include (**a**) Nrf2, (**b**) HO-1, and (**c**) NQO-1. Major inflammatory factors include: (**d**) IL-1β and (**e**) TNF-α. Key genes in the MAPK-JNK pathway for apoptosis include: (**f**) Bax/Bcl-2, (**g**) Caspase-3, and (**h**) Caspase-9.

**Figure 4 animals-12-03320-f004:**
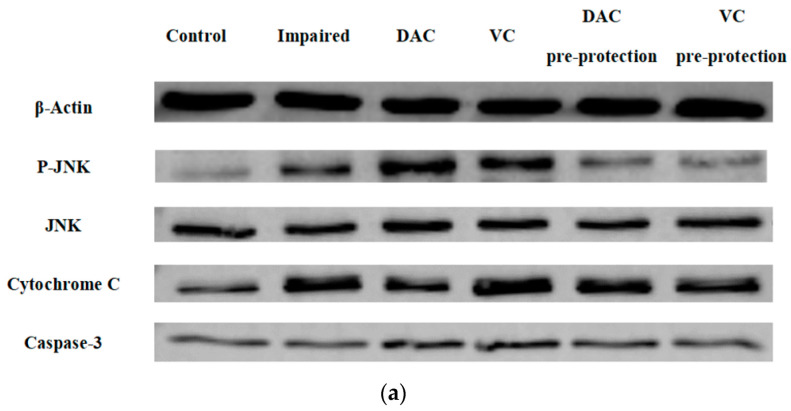
Effect of DAC pre-protection on relative expression of key proteins in MAPK-JNK signaling pathway in GMEC. The band of the western blot is shown in (**a**); the proteins that play key roles in MAPK-JNK pathway are shown in (**b**) P-JNK/JNK, (**c**) Cytochrome-C, and (**d**) Caspase-3.

**Figure 5 animals-12-03320-f005:**
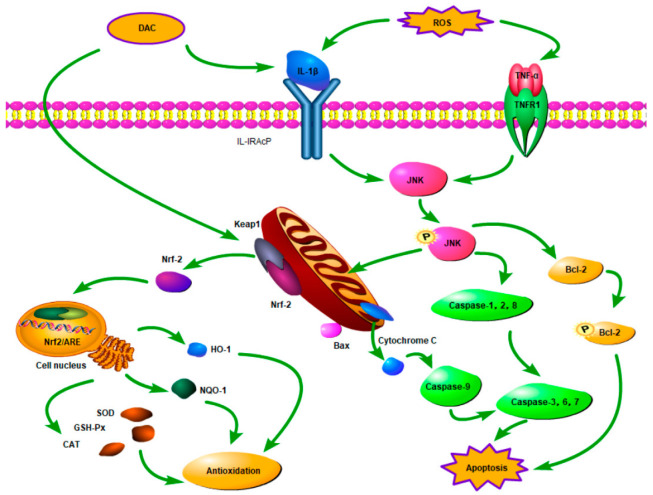
On the basis of above experimental results, the possible mechanisms of GMEC regulating its antioxidant capacity were mapped.

**Table 1 animals-12-03320-t001:** Main nutrients in feed.

Ingredient	Dry Matter (%)					
	Crude Protein (CP) ^1^	Crude Fat (CF) ^2^	Neutral Detergent Fiber (NDF) ^3^	Acid Detergent Fiber (ADF) ^4^	Calcium (Ca) ^5^	Total Phosphorus (TP) ^6^
Corn	7.786	4.722	19.911	2.258	0.200	0.204
Bran	17.966	2.429	47.431	12.992	0.207	0.512
Soybean Meal	45.774	0.784	23.184	14.380	0.559	0.298
*Dioscorea alata* L.	9.550	1.855	25.662	5.540	-	-

^1^ CP content was determined according to the method provided in GB/T 6432-2018; ^2^ CF was according to GB/T 6433-2006; ^3^ NDF was according to GB/T 20806-2006; ^4^ ADF was according to NY/T 1459-2007; ^5^ Ca was according to GB/T 6436-2018; and ^6^ TP was according to GB/T 6437-2018. All standards are set by the Ministry of Agriculture and Rural Affairs of China.

**Table 2 animals-12-03320-t002:** Concentrate formulation.

Item	Content (%) ^1^	
	Control	Treatment
Corn	67	47
Bran	10	11
Soybean Meal	18	17
*Dioscorea alata* L. powder	0	20
Shell powder	1.4	1.4
Sodium bicarbonate	0.6	0.6
CaHPO_4_	1	1
Premix ^2^	1	1
Salt	1	1
Total	100	100
	Dry Matter	
CP %	15.25	15.33
CF %	3.55	2.99
NDF %	22.26	23.65
ADF %	5.40	6.04
Metabolisable energy ^3^ (ME MJ/kg)	13.45	13.43

^1^ Two groups of Hainan black goats were fed different diets (n = 14 each), and the proportions of the components have been measured. ^2^ The premix provides vitamin A 15,000 IU, vitamin D 5000 IU, vitamin E 50 mg, iron 90 mg, copper 12.5 mg, zinc 100 mg, manganese 130 mg, selenium 0.3 mg, and iodine 1.5 mg per kg of diet. ^3^ According to Diao’s monograph [25], ME = 13.670–0.101NDF (R^2^ = 0.880), and Jia also referred to this book in his research on goats [26].

**Table 3 animals-12-03320-t003:** The gene primers of qPCR.

Gene ^1^	Primer Sequence (5′→3′)
Bax	F: TTTCCGACGCACTTCAAC R: CTCGAAGGAAGTCCAATGTC
Bcl-2	F: TCTCCGGCTCACAGCAC R: CAGCCAGGAAATCAACAG
Nrf2	F: TGACAATGAGGTTTCTTCG R: GTGGCTACCTGAACGAACA
NQO-1	F: AGTCCCTGCCATCCTGAA R: TCGGGAGTGTGCCCAATG
Caspase-3	F: CATTATTCAGGCCTGCCGAG R: CTCGAGCTTGTGAGCGTACT
Caspase-9	F: GGGAAATGCTGATCTGGCCT R: CAGCCGTGAGAGAGGATGAC
TNF-α	F: CAAGTAACAAGCCGGTAGCCC R: CCTGAAGAGGACCTGCGAGTAG
HO-1	F: GAACGCAACAAGGAGAAC R: CTGGAGTCGCTGAACATAG
IL-1β	F: CATGTGTGCTGAAGGCTCTC R: AGTGTCGGCGTATCACCTTT
GAPDH	F: CCGTTCGACAGATAGCCGTAA R: AGGATCTCGCTCCTGGAAGA

^1^ All genes can be proofread on the NCBI.

**Table 4 animals-12-03320-t004:** The effects of experimental feed on antioxidant capacity in blood of perinatal female goats.

Group	Item	Time (d)					
		1	2	3	7	14	21
Control	CAT (U/mL)	4.923 ± 0.493 ^Aa^	3.425 ± 0.496 ^Aa^	4.234 ± 0.64 ^Aa^	4.458 ± 0.875 ^Aa^	3.925 ± 0.615 ^Aa^	3.853 ± 0.760 ^Aa^
Treatment		3.295 ± 0.639 ^Aa^	3.495 ± 0.535 ^Aa^	3.597 ± 0.839 ^Aa^	4.351 ± 0.737 ^Aa^	5.030 ± 0.878 ^Aa^	4.760 ± 0.921 ^Aa^
Control	T-AOC (U/mL)	1.406 ± 0.121 ^Aa^	1.857 ± 0.190 ^Aa^	1.575 ± 0.125 ^Aa^	1.663 ± 0.107 ^Aa^	1.486 ± 0.104 ^Aa^	1.472 ± 0.067 ^Aa^
Treatment		2.105 ± 0.091 ^Ab^	2.007 ± 0.137 ^Aa^	2.044 ± 0.083 ^Ab^	2.112 ± 0.090 ^Ab^	2.097 ± 0.121 ^Ab^	2.098 ± 0.167 ^Ab^
Control	SOD (U/mL)	19.819 ± 0.642 ^Aa^	18.785 ± 1.339 ^Aa^	20.477 ± 0.742 ^Aa^	20.372 ± 0.655 ^Aa^	20.300 ± 0.635 ^Aa^	21.579 ± 0.521 ^Aa^
Treatment		25.341 ± 0.426 ^Ab^	24.396 ± 0.373 ^Ab^	25.034 ± 0.282 ^Ab^	24.575 ± 0.914 ^Ab^	24.523 ± 0.738 ^Ab^	24.606 ± 0.746 ^Ab^
Control	GSH-Px (U/mgPr)	466.925 ± 8.353 ^Aa^	483.800 ± 6.671 ^ABa^	487.476 ± 10.872 ^ABa^	500.392 ± 9.243 ^Ba^	497.490 ± 7.727 ^Ba^	491.974 ± 8.668 ^ABa^
Treatment		477.236 ± 9.820 ^Aa^	481.557 ± 7.950 ^Aa^	492.888 ± 12.282 ^ABa^	533.744 ± 12.036 ^Cb^	521.782 ± 10.494 ^BCa^	507.744 ± 13.389 ^ABCa^
Control	MDA (nmol/mL)	11.594 ± 0.684 ^Aa^	11.544 ± 0.662 ^Aa^	9.966 ± 0.343 ^ABa^	8.971 ± 0.862 ^Ba^	9.010 ± 0.569 ^Ba^	11.103 ± 0.937 ^ABa^
Treatment		9.601 ± 0.535 ^Ab^	7.588 ± 0.366 ^Bb^	8.280 ± 0.623 ^Bb^	7.654 ± 0.402 ^Ba^	7.762 ± 0.51 ^Ba^	9.288 ± 0.496 ^Ab^

Compared with the control group, different capital letters of shoulder labels in the same row showed significant difference (*p* < 0.05); there was no significant difference in the same alphabet (*p* > 0.05). Different lowercase letters of shoulder labels in the same column indicate significant difference (*p* < 0.05). The same letter of shoulder label indicates no significant difference (*p* > 0.05), the same as below of the table.

**Table 5 animals-12-03320-t005:** The effects of experimental feed on antioxidant capacity in milk of perinatal female goats.

Group	Item	Time (d)					
		1	2	3	7	14	21
Control	CAT (U/mL)	22.992 ± 2.073 ^Aa^	20.464 ± 1.465 ^Aa^	20.314 ± 1.523 ^Aa^	24.357 ± 1.903 ^Aa^	31.612 ± 1.207 ^Ba^	30.967 ± 1.719 ^Ba^
Treatment		28.128 ± 1.636 ^Aa^	27.762 ± 2.145 ^Ab^	30.620 ± 2.420 ^Ab^	25.680 ± 2.163 ^Aa^	30.389 ± 1.556 ^Aa^	38.509 ± 0.800 ^Bb^
Control	T-AOC (U/mL)	38.262 ± 2.618 ^ABa^	41.295 ± 2.564 ^Ba^	33.390 ± 2.543 ^ABa^	50.853 ± 3.200 ^Ca^	38.171 ± 3.760 ^ABa^	32.049 ± 2.091 ^Aa^
Treatment		42.584 ± 1.88 ^Aa^	57.446 ± 3.639 ^Bb^	54.911 ± 3.753 ^Bb^	38.368 ± 3.314 ^Ab^	44.634 ± 3.994 ^Aa^	44.489 ± 2.927 ^Ab^
Control	SOD (U/mL)	9.963 ± 1.481 ^ABa^	11.019 ± 1.717 ^Ba^	12.053 ± 0.896 ^Ba^	11.342 ± 1.622 ^Ba^	6.176 ± 1.035^Aa^	7.917 ± 0.954 ^ABa^
Treatment		12.197 ± 1.358 ^Aa^	9.776 ± 1.918 ^Aa^	11.693 ± 1.882 ^Aa^	9.457 ± 1.496 ^Aa^	12.06 ± 1.873 ^Ab^	9.581 ±1.374 ^Aa^
Control	GSH-Px (U/mgPr)	478.876 ± 6.981 ^Aa^	490.966 ± 6.037 ^Aa^	481.148 ± 7.832 ^Aa^	480.891 ± 9.380 ^Aa^	492.499 ± 12.768 ^Aa^	491.825 ± 8.849 ^Aa^
Treatment		495.336 ± 9.497 ^Aa^	508.435 ± 10.544 ^Aa^	501.153 ± 9.089 ^Aa^	517.455 ± 5.797 ^Ab^	513.965 ± 7.843 ^Aa^	512.936 ± 7.321 ^Aa^
Control	MDA (nmol/mL)	12.015 ± 1.868 ^Aa^	10.761 ± 1.513 ^ABa^	8.417 ± 0.788 ^ABa^	7.449 ± 1.114 ^Ba^	8.83 ± 0.987 ^ABa^	10.393 ± 0.806 ^ABa^
Treatment		8.267 ± 0.725 ^Ab^	7.956 ± 0.801 ^Aa^	6.515 ± 0.583 ^Aa^	6.270 ± 0.728 ^Aa^	7.545 ±0.497 ^Aa^	7.239 ± 0.654 ^Ab^

**Table 6 animals-12-03320-t006:** The effects of experimental feed on fermentation parameters in rumen of perinatal female goats.

Item	Group			
	Control		Treatment	
	Experiment 14 d	Postpartum 21 d	Experiment 14 d	Postpartum 21 d
Acetic acid (mmol/L)	32.730 ± 1.921 ^A^	28.663 ± 2.395 ^A^	29.887 ± 0.015 ^A^	28.208 ± 1.140 ^A^
Propionic acid (mmol/L)	10.807 ± 0.544 ^A^	8.609 ± 0.322 ^B^	9.096 ± 0.003 ^B^	7.141 ± 0.213 ^C^
Acetic acid/Propionic acid	3.043 ± 0.218 ^A^	3.325 ± 0.215 ^A^	3.286 ± 0.001 ^A^	3.944 ± 0.067 ^B^
Butyric acid (mmol/L)	6.231 ± 0.235 ^A^	5.246 ± 0.312 ^B^	4.246 ± 0.096 ^C^	4.080 ± 0.080 ^C^
TVFA (mmol/L)	53.043 ± 1.889 ^A^	44.843 ± 2.332 ^B^	45.996 ± 0.143 ^B^	41.567 ± 1.336 ^B^
pH	6.84 ± 0.12 ^AB^	6.60 ± 0.06 ^A^	7.07 ± 0.01 ^BC^	7.20 ± 0.07 ^C^

**Table 7 animals-12-03320-t007:** The effects of experimental feed on antioxidant capacity in blood of newborn kids.

Group	Item	Time (d)		
		7	14	21
Control	CAT (U/mL)	3.046 ± 0.470 ^ABa^	2.501 ± 0.280 ^Aa^	3.648 ± 0.448 ^Ba^
Treatment		2.262 ± 0.079 ^Aa^	4.119 ± 1.340 ^ABa^	7.369 ± 1.451 ^Bb^
Control	T-AOC (U/mL)	3.772 ± 0.302 ^Aa^	4.050 ± 0.362 ^Aa^	3.440 ± 0.276 ^Aa^
Treatment		5.045 ± 0.452 ^Ab^	4.791 ± 0.331 ^Aa^	4.410 ± 0.358 ^Ab^
Control	SOD (U/mL)	16.914 ± 1.771 ^Aa^	18.510 ± 1.978 ^Aa^	20.051 ± 1.145 ^Aa^
Treatment		24.658 ± 0.238 ^Ab^	24.586 ± 0.651 ^Ab^	24.916 ± 0.506 ^Ab^
Control	GSH-Px (U/mgPr)	441.24 ± 13.109 ^Aa^	513.143 ± 16.634 ^Ba^	459.186 ± 7.421 ^Aa^
Treatment		464.522 ± 17.104 ^Aa^	520.154 ± 11.981 ^Ba^	483.484 ± 14.614 ^ABa^
Control	MDA (nmol/mL)	11.180 ± 0.297 ^Aa^	11.124 ± 0.335 ^Aa^	10.786 ± 0.488 ^Aa^
Treatment		9.801 ± 0.417 ^Ab^	9.327 ± 0.370 ^Ab^	9.731 ± 0.364 ^Aa^

**Table 8 animals-12-03320-t008:** The effect of DAC on relative survival rate of GMEC (%).

Concentration of DAC (μg/mL)	Treatment Time (h)			
	3	5	7	9
0	100.0 ± 0 ^Aa^	100.0 ± 0 ^Ab^	100.0 ± 0 ^Aa^	100.0 ± 0 ^Aa^
15	99.5 ± 2.3 ^Aa^	104.4 ± 0.9 ^ABbc^	106.8 ± 0.6 ^ABab^	110.7 ± 3.7 ^Bbc^
20	100.2 ± 1.9 ^Aa^	106.9 ± 0.3 ^ABcd^	105.9 ± 1.2 ^Aab^	114.9 ± 4.6 ^Bc^
40	101.7 ± 2.2 ^Aa^	110.1 ± 1.7 ^Bcd^	109.6 ± 2.6 ^Bbc^	114.0 ± 1.3 ^Bc^
50	101.3 ± 2.3 ^Aa^	106.8 ± 3.4 ^ABcd^	118.8 ± 2.9 ^Bd^	110.6 ± 5.1 ^ABbc^
70	95.2 ± 3.7 ^Ab^	92.5 ± 2.0 ^Aa^	103.4 ± 2.7 ^Bab^	103.1 ± 2.5 ^Bab^
100	110.1 ± 3.1 ^Ab^	111.9 ± 1.5 ^Ad^	117.0 ± 4.9 ^Acd^	114.9 ± 0.8 ^Ac^

**Table 9 animals-12-03320-t009:** The effect of DAC on LDH activity in GMEC (U/L).

Concentration of DAC (μg/mL)	Treatment Time (h)			
	3	5	7	9
0	189.217 ± 6.799 ^Aa^	176.775 ± 7.477 ^Aa^	188.803 ± 8.776 ^Aa^	197.201 ± 11.667 ^Aa^
15	196.475 ± 13.716 ^Aa^	180.923 ± 9.891 ^Aab^	190.254 ± 9.215 ^Aa^	209.953 ± 10.775 ^Aab^
20	198.341 ± 17.226 ^Aa^	185.070 ± 11.776 ^Aab^	205.806 ± 10.825 ^Aab^	217.937 ± 8.019 ^Aab^
40	219.907 ± 6.652 ^Aa^	199.585 ± 4.519 ^Aab^	218.662 ± 14.380 ^Aab^	206.843 ± 8.979 ^Aab^
50	214.101 ± 4.519 ^Aba^	191.291 ± 7.183 ^Aab^	224.469 ± 8.098 ^Bab^	223.432 ± 9.890 ^Bab^
100	219.284 ± 6.475 ^Aa^	207.880 ± 6.307 ^Ab^	231.726 ± 11.215 ^Ab^	235.874 ± 12.613 ^Ab^

**Table 10 animals-12-03320-t010:** The effect of DAC on relative content of ROS in GMEC (%).

Concentration of DAC (μg/mL)	Treatment Time (h)			
	3	5	7	9
0	100.0 ± 0 ^Aa^	100.0 ± 0 ^Aa^	100.0 ± 0 ^Aa^	100.0 ± 0 ^Aa^
15	109.4 ± 2.2 ^Ab^	91.2 ± 1.8 ^Bb^	88.6 ± 0.015 ^Bb^	85.4 ± 1.8 ^Bb^
20	100.2 ± 3.1 ^Aa^	89.5 ± 2.9 ^Bb^	82.1 ± 0.027 ^BCc^	80.0 ± 1.4 ^Cc^
40	95.3 ± 1.8 ^Aa^	87.3 ± 1.4 ^Bb^	81.8 ±0.016 ^Cc^	79.0 ± 0.5 ^Cc^
50	96.3 ± 4.3 ^Aa^	86.7 ± 0.9 ^Bb^	79.6 ± 0.005 ^BCc^	78.7 ± 0.4 ^Cc^
100	98.0 ± 2.8 ^Aa^	85.7 ± 2.1 ^Bb^	79.3 ± 0.025 ^Bc^	78.5 ± 1.9 ^Bc^

**Table 11 animals-12-03320-t011:** The effect of DAC on CAT activity in GMEC (U/mL).

Concentration of DAC (μg/mL)	Treatment Time (h)			
	3	5	7	9
0	1.295 ± 0.299 ^Aa^	1.393 ± 0.370 ^Aa^	1.540 ± 0.243 ^Aa^	1.753 ± 0.206 ^Aa^
15	0.656 ± 0.158 ^Aa^	1.131 ± 0.295 ^ABa^	1.654 ± 0.394 ^BCa^	2.211 ± 0.311 ^Cab^
20	1.049 ± 0.245 ^Aa^	1.114 ± 0.361 ^Aa^	2.653 ± 0.331 ^Bb^	4.257 ± 0.220 ^Cc^
40	0.967 ± 0.286 ^Aa^	1.344 ± 0.232 ^ABa^	1.851 ± 0.266 ^ABab^	2.604 ± 0.741 ^Bab^
50	1.311 ± 0.333 ^Aa^	1.900 ± 0.243 ^Aab^	1.933 ± 0.384 ^Aab^	2.031 ± 0.378 ^Aa^
100	0.771 ± 0.279 ^Aa^	2.604 ± 0.612 ^Bb^	2.457 ± 0.204 ^Bab^	3.307 ± 0.327 ^Bbc^

**Table 12 animals-12-03320-t012:** The effect of DAC on T-AOC in GMEC (U/mL).

Concentration of DAC (μg/mL)	Treatment Time (h)			
	3	5	7	9
0	2.025 ± 0.193 ^Aa^	2.019 ± 0.069 ^Aa^	2.434 ± 0.082 ^ABa^	2.864 ± 0.213 ^Ba^
15	2.163 ± 0.118 ^Aab^	2.578 ± 0.084 ^Ab^	3.381 ± 0.269 ^Bb^	3.679 ± 0.311 ^Bb^
20	2.650 ± 0.100 ^Ab^	3.022 ± 0.155 ^Ab^	3.928 ± 0.170 ^Bc^	4.119 ± 0.186 ^Bb^
40	3.332 ± 0.136 ^Ac^	4.370 ± 0.270 ^Bc^	5.065 ± 0.156 ^Cd^	6.243 ± 0.247 ^Dc^
50	4.426 ± 0.295 ^Ad^	4.590 ± 0.155 ^ABc^	5.371 ± 0.196 ^BCd^	5.969 ± 0.385 ^Cc^
100	5.943 ± 0.179 ^Ae^	6.876 ± 0.267 ^Bd^	7.801 ± 0.181 ^Ce^	8.128 ± 0.104 ^Cd^

**Table 13 animals-12-03320-t013:** The effect of DAC on SOD activity in GMEC (U/mL).

Concentration of DAC (μg/mL)	Treatment Time (h)			
	3	5	7	9
0	7.775 ± 0.347 ^Aa^	7.907 ± 0.345 ^Aa^	7.743 ± 0.364 ^Aa^	7.759 ± 0.430 ^Aa^
15	6.493 ± 0.475 ^Aabc^	9.501 ± 0.253 ^Bb^	17.475 ± 0.353 ^Cb^	16.847 ± 0.533 ^Cb^
20	7.184 ± 0.694 ^Aab^	9.913 ± 0.765 ^Bb^	18.830 ± 0.674 ^Cb^	19.617 ± 0.377 ^Cc^
40	6.099 ± 0.261 ^Abc^	12.099 ± 0.349 ^Bc^	14.882 ± 0.485 ^Cc^	13.655 ± 0.844 ^BCd^
50	6.559 ± 0.350 ^Aabc^	10.373 ± 0.461 ^Bb^	12.373 ± 0.625 ^Cd^	10.795 ± 0.717 ^BCe^
100	5.605 ± 0.381 ^Ac^	10.17 ± 0.705 ^Bb^	10.871 ± 0.237 ^Be^	11.589 ± 0.579 ^Be^

**Table 14 animals-12-03320-t014:** The effect of DAC on GSH-Px activity in GMEC (U/mgPr).

Concentration of DAC (μg/mL)	Treatment Time (h)			
	3	5	7	9
0	238.87 ± 23.88 ^Aa^	175.238 ± 31.053 ^Aa^	172.209 ± 12.746 ^Aa^	225.942 ± 4.287 ^Aa^
15	342.397 ± 22.793 ^Bbc^	249.98 ± 19.244 ^Aab^	296.441 ± 34.218A ^Bb^	462.589 ± 24.024 ^Cc^
20	295.936 ± 5.824 ^Aab^	256.697 ± 23.14 ^Aab^	368.657 ± 11.549 ^Bb^	499.960 ± 17.862 ^Cc^
40	382.798 ± 10.936 ^ABc^	352.497 ± 34.596 ^Ac^	363.607 ± 22.489 ^Ab^	455.014 ± 18.869 ^Bc^
50	338.357 ± 37.889 ^ABbc^	262.101 ± 23.487 ^Aab^	328.257 ± 30.719 ^ABb^	376.232 ± 11.006 ^Bb^
100	276.241 ± 15.874 ^Aab^	301.996 ± 27.443 ^Abc^	365.123 ± 42.994 ^Ab^	334.822 ± 33.753 ^Ab^

**Table 15 animals-12-03320-t015:** The effect of DAC on content of MDA in GMEC (nmol/mL).

Concentration of DAC (μg/mL)	Treatment Time (h)			
	3	5	7	9
0	0.258 ± 0.021 ^Aa^	0.262 ± 0.018 ^Aa^	0.245 ± 0.024 ^Aa^	0.224 ± 0.024 ^Aa^
15	0.245 ± 0.129 ^Aa^	0.207 ± 0.026 ^Aa^	0.180 ± 0.030 ^Aab^	0.167 ± 0.012 ^Ab^
20	0.230 ± 0.007 ^Aa^	0.183 ± 0.092 ^Aa^	0.166 ± 0.043 ^Aab^	0.141 ± 0.010 ^Abc^
40	0.231 ± 0.014 ^Aa^	0.190 ± 0.017 ^Ba^	0.174 ± 0.006 ^Bab^	0.158 ± 0.010 ^Bb^
50	0.240 ± 0.041 ^Aa^	0.173 ± 0.047 ^ABa^	0.139 ± 0.032 ^ABb^	0.112 ± 0.009 ^Bc^
100	0.248 ± 0.058 ^Aa^	0.177 ± 0.029 ^ABa^	0.143 ± 0.021 ^ABb^	0.101 ± 0.004 ^Bc^

**Table 16 animals-12-03320-t016:** The main effect and mutual effect of treatment time and concentration on each index.

Index	Significance of Intersubjective Effect (*p*-Value) ^1^		
	Time	Concentration	Time × Concentration
Survival rate	0.001	0.001	0.006
ROS	0.001	0.001	0.001
LDH	0.001	0.001	0.001
CAT	0.001	0.047	0.099
T-AOC	0.001	0.072	0.229
SOD	0.001	0.001	0.001
MDA	0.097	0.017	0.718
GSH-Px	0.001	0.001	0.005

^1^ Given that two variables were involved in this experiment, the main effects analysis was conducted by SPSS 24 to explore whether the effect of the variable was significant.

## Data Availability

All raw data in the current study are available from the corresponding author on reasonable request.

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
