# Peer review of "The Effect of Anthocyanins from Dioscorea alata L. on Antioxidant Properties of Perinatal Hainan Black Goats and Its Possible Mechanism in the Mammary Gland"

_animals, 2022, doi:10.3390/ani12233320_

Round 1

Reviewer 1 Report (Previous Reviewer 2)

Thank you for your revision and your answers at my comments. Some efforts are significant which led to an improvment of the maunscript. However, some others comments, which are for me very important, have been neglected :

1. The revision of english language is very limited at a few added words and/or sentences (lines 8; 39; 41; 59...) that further affected the quality of the article.

2. In Material and Methods (MM) : some details about animals have been reported but as mean (age, body weight, parity) without standard error which is very important. Moreover, the BCS was given based on the Saanen goats ??? I prefer that you use your data otherwise not worth it. As I asked in my comments, I still ignore the number of kids in each group (is it balanced ? in other words, are the fertility and prolificacy the same on the two groups ?)

3. You have presented results relating to rumen fermentation (TVFA, C2; C3 and C4; pH) that you have not described before in MM.

4. Figure showed a  regression equation between VC concentration and DPPH clearance rate; you must indicate this method in the statistical analysis. Is the interaction between concentration of DAC (n = 6) and treatmant time (n = 4) has been tested (from table 8 to table 15) ? 

5. The titles of all the tables are not well explicit et should more explained the results reported in each table? 

6. Some related results can be regrouped in the same table (antioxidant capacity in blood and milk for goats and/or kids) to try to highlight links between these parameters.

Author Response

Reviewer 2 Report (New Reviewer)

The manuscript needs extensive revision for language and grammar. The text is sometimes confusing and makes the manuscript difficult to understand. All abbreviations must be previously written in full in the text. The abstract must follow the journal's rules.

Author Response

Reviewer 3 Report (New Reviewer)

It is quite a significant experiment, with extensive research on the topic studied. However, it will increase its value if it improves the details of some methodologies, also the English paper's writing.

Author Response

Reviewer 4 Report (New Reviewer)

Anthocyanins possess a potent ability to act as free radical scavengers against reactive oxygen species. This manuscript describes the effects of Dioscorea alata L. in antioxidant capacity in perinatal female goats and newborn kids. Meanwhile, author studied the antioxidative effects of Dioscorea alata L. anthocyanins (DAC) on GEMC and related mechanisms. The author conducted a tremendous hard work on animal studies, cell culture, as well as molecular biology experiments. However, the manuscript has some major flaw as follow.

1. The author didn’t provide enough results to support the idea that Dioscorea alata L. has the antioxidant effect on mammary gland in vivo. Without testing the ROS status directly from goat mammary gland tissue, only the data of lower ROS in milk and newborn kids can’t support rationale to conduct the following DAC co-culture GMEC study ex vivo.

2. Line 128, according bncc.org.cn website, they DOES NOT sell any commercial cell-line of mammary epithelial cells from goat. The author should provide more detail information, including the catalog number, in the manuscript, in order to help other researcher to follow up your study.

3. The experiment took days to study the function of Dioscorea alata L. on goats, why the author only designed the DAC treatment based on 9 hours maximum? The results of the acute studies could NOT provide sufficient evidence of molecular mechanism to explain the chronic effects of Dioscorea alata L. on goats.

Some minor issues,

4. Typos, CCK (line 140) or CCk (line 78), Nrf2 (413) or Nrf-2 (439), dd-H2O (line 79), et al.

5. Please spend more time to polish the format of the figures, such as black color the x- and y-axis for Figure 1.

6. Table 4 and 5, why not show the antioxidant status in blood and milk samples from day 0 or before the treatment? These data could provide sufficient evidence to resolve the group bias, because it is really amazing that the Dioscorea alata L. supplement takes one day to in effect on antioxidant capacity in goats.

Round 2

Reviewer 2 Report (New Reviewer)

I consider that once all the changes suggested by the reviewers have been made, the manuscript is ready for publication.

Author Response

Thank you for your valuable guidance during this period.

Reviewer 4 Report (New Reviewer)

It looks like the author answered and fixed most of questions based on previous review. However there are still some format errors in the manuscript. For example,

Line 279, correct space before R square;

Line 311, 303, Nrf2 NRF2 Nrf-2 format needed be consisted throughout the paper;

Line 605, H2O2.

Author Response

Thank you for your careful review and valuable advice during this period. I have fixed the formatting error that you have put forward。

This manuscript is a resubmission of an earlier submission. The following is a list of the peer review reports and author responses from that submission.

Round 1

Reviewer 1 Report

Dear authors,

Please see below comments on the paper submitted to the journal animals. In general, the idea of the work is interesting, but it has serious description problems in materials and methods, statistical analysis and presentation, and discussion of the results. With all respect, the text is very unattractive to the reader, repetitive, with grammatical problems, and not very technical in some parts. I recommend a thorough review of the same so that it can be submitted again.

Overall comments:

1)      The English need to be improved. Several sentences are too long and confusing (for instance, lines 8, 9, and 10 – simple summary). In the results section, each sub-section was very repetitive, with a lack of standardization in formatting, which makes the text very tiring to read. Furthermore, the tables presented could be better formatted or even more summarized to allow the reader a better understanding of the content of the paper.

2)      It was not clear whether the work was carried out using goats or sheep as an animal model. In several parts of the text, the words goats, sheep, and ewes are used interchangeably, which makes no sense (look at lines 92, 94, 95, 96, 98, and 111).

3)      In general, terms related to the field of animal nutrition were not very technical. For example, in tables 1 and 2, ingredients were identified as materials. Furthermore, NDF and ADF cannot be considered nutrients, as they are feed components defined by their respective methods of analysis. Additionally, these fractions should be corrected at least for the ash content of the ingredients. In the materials and methods section, there is no description of which methods were used to obtain the proximate chemical composition of the feeds used in the study.

Specific comments:

1)      More details about the animals should be provided: what is the mean body weight? what is the mean body condition score? And Parity?

2)      More details about the feed analysis (methods) should be provided. How CP, NDF, ADF were obtained? What does CF stand for? Did you follow the AOAC recommendations for feed analyses? In my opinion, citing a monograph with reference to the calculation of EM is not a good practice. I would recommend using a peer-reviewed paper as a reference (line 109).

3)      Statistical analysis: for comparisons between groups, ANOVA would be enough, without the need for the T-test, since there are only two treatments. Furthermore, measurements taken on the same experimental units at different times should be analyzed as measurements repeated over time (within groups). The description of the statistical model is very important.

Reviewer 2 Report

The work is very intersting since it aims to evaluate anthocyanins from Dioscorea alata L on antioxydant capacity and properties of perinatal gotas by measuring many paramters. The appraoch and methodolgy used in this work allow to achieve the objetctives initally fixed for the trial. However, some comments can be mentionned :

The introduction is limited in some very aspects treated in this work as, for example, the differences among anthocyanins should be indicatedthat since they are the most interesting and vigorously studied plant compounds. the other uses of these anthocyanins are hidden. This introduction highlights the effect of the state of Negative Energy Balance (NEB), directly and/or indirectly, on lipid peroxydation, ROS production and thus on damage leading to the stress situation. However, this situation is unclear in your MM.

The MM should be greatly improved, you have to indicate all the informations about animals (age, body weight notably loss or gain, body score, prolicacy, fertility, youth mortality....); replace ewes by goats, and lambs by kids; the number of animals changed ( twenty-eight goats, n=10 for each group ?, the difference ?; amounts of feeds distrubuted to goats and part of concentrate ? subsitution of corn 20 % of concentrate and 30 % of corn.

The statistical analysis, results are well presented and make it possible to achieve the objectives initially assigned to the work. However, the discussion of the results shoulb be improved for some aspects especially the effects on rumen fermentation in relation with the nature of substitution (although the two diets re iso-energetic and iso-nitrogen). relation between starch (if DAC contains more starch as you cited) and C3 and pH (lines 375 and 376) ? relation between anthocyanins and protein fermentation and metabolism should be more explained (lines 377-379) Again, you integretad the NEB in this discussion, but this parameter was not measured in this work.  Finally, the compilation of the effects of all these parameters on the discussion of overall improvment of antioxydant capacity of the cells of animals supplemented by DAC, should better refined.

The conclusion related the main results of the work and the references are well cited and are in relation with the subject.

Author Response

请参阅附件。
